# Wood Modification Using Imidazole and Succinimide: Effects on Dimensional Stability and Bending Properties

**Alexander Scharf** [1,*], **Henric Dernegård** [2], **Johan Oja** [3], **Dick Sandberg** [1] and **Dennis Jones** [1]

1 Wood Science and Engineering, Luleå University of Technology, Forskargatan 1, SE-93187 Skellefteå, Sweden; dick.sandberg@ltu.se (D.S.); dennis.jones@ltu.se (D.J.)
2 Holmen AB, Strandvägen 1, SE-11451 Stockholm, Sweden; henric.dernegard@holmen.com
3 Norra Timber, Skeppargatan 1, SE-90403 Umeå, Sweden; johan.oja@norraskog.se
* Correspondence: alexander.scharf@ltu.se; Tel.: +46-730825153

**Abstract:** The modification of Scots pine sapwood (*Pinus sylvestris* L.) with the heterocyclic compounds imidazole and succinimide was investigated. Pressure-impregnation with aqueous solutions containing imidazole, imidazole + citric acid, succinimide, succinimide + citric acid, and citric acid + sorbitol (CIOL®) with solid contents of 5%, 10%, and 15% was followed by oven-curing at 220 °C for 1 h. During the treatment steps, the changes in mass, bending properties, and anti-swelling efficiency (ASE) were examined. The results indicate that solid concentrations within the range of 5% to 10% were optimal. The results seem to show that there are two differing mechanisms in the modification of imidazole and succinimide, respectively. Mass loss due to heat treatment was highest in the imidazole-treated specimens, whereas it remained low and concentration-independent in the succinimide-treated specimens. After three cycles, the ASE reached 31% for the imidazole-treated specimens and improved to 38% with the addition of citric acid. For succinimide, the ASE increased from 17% to 41%. The bending properties generally showed improvement, except for succinimide + citric acid and CIOL®, which displayed a reduced modulus of rupture. Chemical analyses are warranted to fully understand the reaction mechanisms of these treatments. The positive effects of imidazole treatment are suggested to stem from a thermal reaction between the chemical and the wood, indicated by substantial mass loss during leaching and specimen darkening. Succinimide and citric acid might exhibit polymerization with each other and with wood components, which is akin to the CIOL® process. Further research should delve into the reaction mechanisms and the impact of imidazole and succinimide on biological durability.

**Keywords:** anti-swelling efficiency; citric acid; mechanical properties; thermal treatment; wood protection





## 1. Introduction

Wood modification involves altering the structural and chemical composition of the wood cell wall using heat or chemicals, and this is primarily aimed at enhancing water-related characteristics like sorption, dimensional stability, and resistance to biological degradation [1,2].

Thermal wood modification, achieved through heat treatment in low-oxygen environments within the temperature range of 160 °C to 240 °C, induces favorable changes in the chemical composition of the cell wall [3]. The central mechanism driving alterations in the cell wall is the hydrolysis-induced degradation of hemicelluloses, yielding commonly observed byproducts such as furfural and hydroxymethylfurfural [4–6]. Lignin degradation also occurs at temperatures exceeding 220 °C [7]. Condensation reactions of oxidatively cleaved ether linkages can result in carbonyl and phenolic groups, which lead to the formation of new crosslinks [8]. The formation of various degradation products accounts for a mass loss of up to 20%. However, not all of the degradation products are

volatiles, and thus, they partly remain in the wood until extracted by a solvent such as water [7]. The positive impact of heat treatment on sorption and dimensional stability arises from the reduced presence of moisture-sensitive hydroxyl groups within hemicelluloses [9], coupled with the generation of novel crosslinks facilitated by degradation products [10]. Furthermore, the lowered availability of accessible sugars and decreased equilibrium moisture content contribute to an enhanced level of biological durability [11]. The strength properties of thermally modified wood are negatively affected due to the reduced degree of polymerization of the holocelluloses [12], as is commonly exhibited during bending tests with a reduced modulus of rupture (MOR), often accompanied by a brittle failure mode. Both biological durability and a decrease in bending properties correlate to mass loss during thermal modification [13].

Chemical wood modification is based on the introduction of various chemicals into the wood. It exists in different forms with the main difference being the location of chemical deposition and the type of bonding with the cell wall [1]. Chemicals can react with groups of cell-wall polymers, blocking, e.g., hydroxyl groups, or leading to crosslinking by reacting with two hydroxyl groups. The former results in cell-wall bulking, whereas the latter additionally limits the maximum distance between the cell-wall polymers, effectively reducing the maximum swelling, i.e., providing anti-swelling efficiency (ASE) [14]. Additionally, lumen-filling treatments exists which may or may not react with the cell-wall components. Similar to thermal modification, chemical modification improves water-related properties and biological durability and can affect mechanical properties [15].

Abundant hydroxyl groups in hemicelluloses drive common esterification reactions for wood modification. Acids like acetic anhydride, succinic or maleic anhydride, polycarboxylic acids, and isocyanates are used, with acetic anhydride-based acetylation being commercialized [2]. Recently, the polycarboxylic acid citric acid gained attention due to its cost-effective availability from microbial fermentation using *Aspergillus niger* [16]. Initially applied in the cotton industry [17], polycarboxylic acid was later adapted for wood modification [18,19]. The reaction mechanism between wood and citric acid involves a two-step esterification process, where a cyclic anhydride is initially formed, followed by its reaction with hydroxyl groups in the wood to create ester linkages [20]. Wood modification with citric acid offers several advantages, such as reduced water absorption, enhanced resistance against termites and fungi, an improved modulus of elasticity (MOE), improved compression strength, and better dimensional stability [21,22]. However, it is important to note that there are some disadvantages, including a strong reduction in MOR, increased brittleness, and a yellowing of the treated wood [23]. Feng et al. [24] reported an initial ASE of 48% at a chemical load of 36%, whereas L'Hostis et al. [25] reported an initial ASE of 67% and a low leaching rate. However, the MOR decreased significantly. In order to improve the performance of citric acid-treated wood, compounds containing alcoholic hydroxyl groups leading to polymerization and increased chemical fixation can be incorporated [25–31].

The polymerization of citric acid and sorbitol in an aqueous solution was initially demonstrated by Centolella and Razor [32]. Doll et al. [33] proposed the formation of intermediates and, subsequently, a citrate sorbitol ester, as shown in Figure 1. Larnøy et al. [30] reported the polyesterification of citric acid and sorbitol in a 3:1 molar ratio for solid wood at 140 °C for 18 h at a weight-percentage gain (WPG) of 80%. Leaching tests indicated excellent leaching and fungal resistance, as higher curing temperatures led to higher conversion rates of the functional groups, resulting in a denser crosslinked network in the cell wall. Mubarok et al. [34] reported an ASE of 55% at a WPG of 30%. However, similar to citric acid treatments, the MOR and work-to-maximum load in bending decreased significantly. Beck [35] showed an ASE of 40% at WPGs of 14%–31%, with leaching rates below 2%. For more detailed information on citric acid in wood modification, readers are referred to the review by Lee et al. [23].

**Figure 1.** The reaction mechanism between citric acid (**A**) and sorbitol (**C**). Intermediate cyclic anhydride (**B,E**), ester (**D**), and anhydrosorbitol ring (**F**). Adapted from [30].

Imidazole, a heterocyclic compound comprising two nitrogen and three carbon atoms, possesses acidic and basic properties. It is environmentally benign and nonhazardous. Morais et al. [36] pioneered the use of imidazole for the extraction of cellulose and hemicelluloses from native wheat straw at a temperature of 170 °C. The depolymerization process yielded a range of valuable lignin-derived compounds, including vanillin, vanillic acid, and rosmarinic acid. Since then, extensive research efforts have been directed toward exploring the potential of imidazole as a solvent in the field of biomass valorization. Grylewicz et al. [37] investigated the use of imidazole in combination with glycerol for the fabrication of thermoplastic starch and wood-fiber composites, reporting enhanced moisture sorption and surface hydrophobicity properties. Del Menezzi et al. [38] reported that citric acid not only reacted with the hydroxyl groups of hemicelluloses but also that the reaction with the aliphatic chains of lignin was possible. It is, hence, possible that a combined treatment of lignin-altering imidazole and citric acid might lead to enhanced crosslinking in the wood.

Succinimide, another heterocyclic compound containing nitrogen, exhibits a carbonyl and an amide group. Succinimides have high chemical reactivity due to the presence of both the carbonyl and methylene groups [39]. It has applications in the pharmaceutical, polymer, and material industries [40]. Under certain conditions, the carbonyl group of succinimide can undergo nucleophilic additions with hydroxyl groups leading to the formation of ester linkages. The reaction is typically catalyzed by acidic conditions.

To the best of our knowledge, no previous studies have explored the application of imidazole or succinimide in wood modification. Thus, our research aimed to explore wood modification systems involving imidazole and succinimide alone and in combination with citric acid. The treatment was based on pressure impregnation and subsequent heat treatment to improve the mechanical and hygroscopic properties.

## 2. Materials and Methods

### 2.1. Materials

Scots pine (*Pinus sylvestris* L.) sawn timber harvested in Västerbotten County was obtained from a sawmill in northern Sweden (Norra Timber, Kåge, Sweden). Scots pine is

an abundant and economically important species in northern Europe and relatively easy to impregnate. The timber was industrially dried to a moisture content (MC) of approx. 18%.

The chemicals used in this study were synthesis-grade >98% imidazole ($C_3N_2H_4$) powder (IoLiTec-Ionic Liquids Technologies GmbH, Heilbronn, Germany), synthesis-grade >98% succinimide ($C_4H_5NO_2$) (Sigma-Aldrich, Merck KGaA, Darmstadt, Germany), 99.9% analytical-grade citric acid ($C_6H_8O_7$), and ≥96% technical-grade D (-)-sorbitol ($C_6H_{14}O_6$) powder (VWR International AB, Stockholm, Sweden).

### 2.2. Specimen and Solution Preparation

Visually selected, defect-free, straight-grained sapwood specimens were prepared from the sawn timber for further pressure impregnation and heat treatment and the subsequent determination of anti-swelling efficiency (ASE) and bending properties. The specimens used for ASE test had dimensions of 21 × 20 × 10 mm (radial × tangential × longitudinal), and the specimens for the bending test had dimensions of 10 × 10 × 200 mm (radial × tangential × longitudinal). The specimens were prepared in surplus and were conditioned for 4 weeks at 20 °C and 65% relative humidity until they had a constant mass and, thus, the equilibrium moisture content (EMC) was reached. For both tests, the specimens were grouped by density, and the lowest and highest 10% were sorted out. The remaining specimens exhibited a mean density at an EMC of 480 ± 23 kg/m$^3$. They were randomly distributed based on density into 17 groups. Additionally, the cross-sections of the bending specimens were end-sealed with silicon, preventing uneven liquid flow during treatment, and thus securing even modification throughout the specimens.

A total of five different chemical combinations were dissolved in deionized water to produce solutions with 5, 10, and 15 wt% solid contents. The chemical combinations are shown in Table 1. The treatment CIOL® was used as a reference representing a bio-based wood modification process, which is close to commercialization [30,41,42]. Additionally, a solely heat-treated (H) and untreated control (C) group were added.

**Table 1.** Treatments and chemical concentration of aqueous solutions used for pressure impregnation. The number in the treatment ID states the solid concentration of the used solution. Number of specimens per group: anti-swelling efficiency test = 6, and bending test = 12.

| Treatment ID | Total Concentration of Solution (wt%) | Imidazole (wt%) | Succinimide (wt%) | Citric Acid (wt%) | Sorbitol (wt%) | Heat Treatment Temperature (°C) |
|---|---|---|---|---|---|---|
| I5 | 5 | 5.0 | - | - | - | 220 |
| I10 | 10 | 10.0 | - | - | - | 220 |
| I15 | 15 | 15.0 | - | - | - | 220 |
| ICA5 | 5 | 2.8 | - | 2.2 | - | 220 |
| ICA10 | 10 | 5.6 | - | 4.4 | - | 220 |
| ICA15 | 15 | 8.4 | - | 6.6 | - | 220 |
| Su5 | 5 | - | 5.0 | - | - | 220 |
| Su10 | 10 | - | 10.0 | - | - | 220 |
| Su15 | 15 | - | 15.0 | - | - | 220 |
| SuCA5 | 5 | - | 2.8 | 2.2 | - | 220 |
| SuCA10 | 10 | - | 5.6 | 4.4 | - | 220 |
| SuCA15 | 15 | - | 8.4 | 6.6 | - | 220 |
| CIOL®5 [1] | 5 | - | - | 3.8 | 1.2 | 220 |
| CIOL®10 [1] | 10 | - | - | 7.6 | 2.4 | 220 |
| CIOL®15 [1] | 15 | - | - | 11.4 | 3.6 | 220 |
| H | - | - | - | - | - | 220 |
| C | - | - | - | - | - | - |

[1] The combination of citric acid and sorbitol in a 3:1 molar ratio fixed by a curing step is carried out according to the CIOL® process [42].

### 2.3. Pressure Impregnation and Heat Treatment

For each group to be chemically treated, the conditioned specimens for ASE and the bending tests were placed together in a vessel and fully submerged with the respective chemical solution. The specimens were pressure-impregnated in an autoclave using a full-cell method, whereby a vacuum at 20 mbar was initially applied for 30 min, followed by 1 h pressure at 15 bar. The excess solution was wiped off with tissue paper, and the mass and dimensions were recorded. This was followed by oven-drying (open system) to 0% MC for 24 h at 70 °C and 16 h at 103 °C. The mass and dimensions of each specimen were recorded to determine solution uptake and WPG, which were calculated as follows (Equations (1) and (2)):

$$Solution\ uptake = \frac{m_1 - m_0}{m_0} - 0.11 \tag{1}$$

$$WPG = (m_2 - m_0)/m_0 \tag{2}$$

where $m_0$ is the initial dry mass, $m_1$ is the mass directly after impregnation, and $m_2$ is the oven-dry mass after impregnation, where $m_0$ was estimated for all specimens based on the EMC of the control group C. The conditioning of the specimens to EMC resulted in an MC of $11.0 \pm 0.27\%$, determined by the gravimetric method. Additionally, the MC of 11% was subtracted from the solution uptake to account for the vapor-bound water as the specimens were impregnated in a conditioned state.

The dried specimens were tightly wrapped in aluminum foil and placed in an oven, undergoing heat treatment in an open system at 220 °C for 1 h. The mass and dimensions were recorded, and the mass loss ($\Delta m_{HT}$) and bulking coefficient (*BC*) after heat treatment were calculated (Equations (3) and (4)):

$$\Delta m_{HT} = (m_3 - m_0)/m_0 \tag{3}$$

$$BC = (V_3 - V_0)/V_0 \tag{4}$$

where $m_3$ and $V_3$ are the mass and wood volume after heat treatment, respectively, and $V_0$ is the initial dry wood volume. Referencing the mass loss during the heat treatment to the initial dry mass $m_0$ instead of $m_2$ was a deliberate choice to account for differences in WPG, providing results that are easier to compare.

The heat-treated specimens were kept for 24 h at room climate before being placed in a conditioning chamber at 20 °C and 65% relative humidity until EMC was reached.

### 2.4. Specimen Characterization

ASE was evaluated using a previously established procedure [43]. All specimens were subjected to repeated wet-dry cycles. The specimens underwent vacuum impregnation in deionized water for 1 h, followed by immersion in water for 72 h. After water treatment, the specimens were air-dried under ambient conditions and subsequently oven-dried for 48 h at 70 °C and 16 h at 103 °C. The mass and dimensions were measured in a wet and dry state. The ASE was calculated following the method described by Stamm [44] (Equation (5)):

$$ASE = (S_2 - S_1)/S_1 \tag{5}$$

where $S_1$ is the untreated volumetric swelling coefficient, and $S_2$ is the treated volumetric swelling coefficient between the water-saturated and dry states. This wet-dry cycle process was repeated three times to wash out any water-soluble compounds affecting the ASE. Crosslinking was evaluated by the volumetric change over the wet-dry cycles and was calculated according to Equation (6) [43]. In order to evaluate the leaching resistance of the treated wood, the mass loss due to leaching over three cycles $\Delta m_{leaching}$ was calculated (Equation (7)). It describes the share of water-soluble compounds in the wood after heat

treatment, which can be either in the introduced chemicals, (thermal) degradation products, or naturally occurring water-soluble extractives. A high $\Delta m_{leaching}$ indicates the low stability of the introduced chemicals and a potential negative environmental impact during the service life of the product. Additionally, the mass loss $\Delta m_{total}$ over the complete treatment and leaching process was calculated (Equation (8)). This is similar to the corrected mass loss proposed by Altgen et al. [45] and, in contrast to $\Delta m_{HT}$, includes the water-soluble compounds created during the heat treatment. If the $\Delta m_{total}$ of treatment is lower than the $\Delta m_{total}$ of the solely heat-treated group, chemicals are likely to be present in the wood after leaching, while a higher value would suggest that (thermal) degradation was promoted by the treatment.

$$Volumetric\ change = (V_n - V_0)/V_0 \tag{6}$$

$$\Delta m_{leaching} = (m_4 - m_3)/m_3 \tag{7}$$

$$\Delta m_{total} = (m_4 - m_0)/m_0 \tag{8}$$

where $V_n$ is the volume in either the oven-dried or wet state in the nth wet-dry cycle, $V_0$ is the initial oven-dried dimension, and $m_4$ is the mass after three wet-dry cycles and oven-drying for 24 h at 70 °C and 16 h at 103 °C.

The MOR and the local modulus of elasticity (MOE) in bending were tested in conjunction with a four-point bending test. Specimens $10 \times 10 \times 200$ mm in size were loaded in a universal testing machine (MTS System Corporation, Eden Prairie, MN, USA) equipped with a 10 kN load cell according to the EN 408 standard [46]. The span was 180 mm, the distance between the load points was 60 mm, and the loading rate was 0.03 mm/s.

In order to assess the statistical significance of differences between the mean values for each group, a one-way analysis of variance (ANOVA) was conducted for each measured property. Following this, a Tukey post-hoc test was performed to identify specific groups that exhibited significant differences. The significance level was set at $\alpha = 0.05$. The results of the post-hoc test are presented using a compact letter display, wherein those groups that were not significantly different from each other were assigned the same letter.

## 3. Results and Discussion

### 3.1. Impregnation and Heat Treatment

The solution uptake was 138%–150% among the groups with no significant difference between the different solid concentrations of the solutions. Figure 2 shows the WPG after impregnation and drying and the mass loss $\Delta m_{HT}$ after the heat treatment determined for the ASE specimens.

The WPG behaved similarly among the different treatment groups when solutions with a 5 and 10% solid concentration were used, and an increase from 5 to 10% solid concentration in the chemical solution led to an approx. doubling in WPG. The increase from 10% to 15%, however, did not translate into a 1.5-fold increase in WPG. It may be suggested that the saturation of available sites, e.g., for hydrogen bonding, is reached at a solid concentration between 10 and 15%, likely being dependent on the molecular structure of the used chemicals. At 10% and 15%, the groups SuCA and CIOL® showed higher WPG compared to the other treatments, whereas, for imidazole, I15 showed no significant increase in uptake compared to I10.

The mass loss, $\Delta m_{HT}$, in Figure 2 comprised multiple degradative processes. Without chemical treatment, $\Delta m_{HT}$ was $2.6 \pm 0.4\%$ due to thermal degradation, primarily of the hemicelluloses [7]. In the chemically treated groups (except Su), $\Delta m_{HT}$ was related to the WPG with a clear difference between the treatments, with and without imidazole. On the one hand, the thermal degradation of the used chemicals likely happened since it had been noted that both imidazole and citric acid exhibit one-stage degradation, peaking at 220 °C in thermogravimetric analyses [47,48], whereas succinimide starts to thermally degrade around 250 °C [49]. On the other hand, the chemical-promoted degradation of

cell wall polymers is possible. The $\Delta m_{HT}$ in the imidazole treatments was partly higher than the respective WPG and was accompanied by a strong darkening of the specimens. This has been assumed to be a result of imidazole promoting the degradation of lignin during the heat treatment, a fact that has been observed in the refinery of biomass when imidazole has been present [36]. The groups treated with only succinimide showed the least mass loss independently of WPG. Additionally, the groups Su10 and Su15 exhibited a thin layer of salt deposits on the surface when the specimens were handled. It is likely that only a few reaction sites between the cell wall polymers and the succinimide existed or that no bonds were formed. The mass loss in the CIOL® can be attributed to the ester linkage of the carboxyl groups of the citric acid to the hydroxyl groups of the wood and the sorbitol, which already occurs at temperatures of 140 °C. In this process, citric acid transforms into the reactive anhydride with the release of water molecules [30,33]. In the groups ICA and SuCA, 56% of the imidazole and succinimide mass was substituted by citric acid. While the $\Delta m_{HT}$ was little effected in the imidazole groups (from I to ICA), and the $\Delta m_{HT}$ in succinimide groups (from Su to SuCA) increased. This can be due to the catalyzed thermal degradation of hemicelluloses and lignin in the presence of acids [50] and by the esterification of citric acid with the cell wall polymers, which involves dehydration during heat treatment. Interestingly, the mass loss was similar between CIOL® and SuCA and between I and ICA. The formation of crosslinks between imidazole, succinimide, citric acid, and the cell-wall polymers or their degradation products is possible, but chemical analyses are necessary.

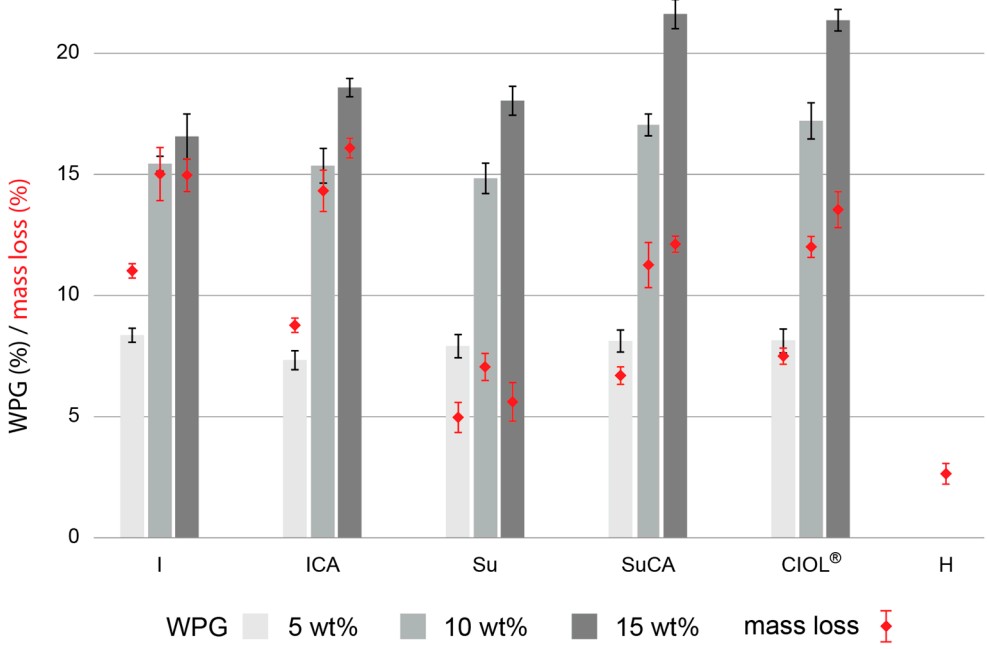

**Figure 2.** The mean ± standard deviation of weight percentage gain (WPG) after pressure impregnation and drying and mass loss due to heat treatment at 220 °C ($\Delta m_{HT}$) of the specimens treated with different concentrations (wt%) of the chemical solution. $\Delta m_{HT}$ was calculated in relation to the initial dry mass. Treatment shorting can be seen in Table 1. The number of specimens per group = 6.

Figure 3 shows the cross-sections of one arbitrarily chosen specimen per group after the heat treatment. A color change was evident in all groups but was the strongest in groups containing imidazole, where the color was similar to carbonized wood. This indicates that imidazole might increase the amount of thermal degradation in wood and potentially lower the charring temperature of wood. Imidazole has been shown to depolymerize lignin in wheat straw, resulting in compounds such as vanillin, vanillic acid, and rosmarinic acid [36]. The groups Su, SuCA, and CIOL® showed slightly different browning compared

to the heat-treated group (H), which could be due to the presence of polymerized chemicals in the wood.

**Figure 3.** Cross-sectional appearance of specimens after heat treatment and the untreated control (C).

### 3.2. Anti-Swelling Efficiency and Water Stability

Table 2 shows the EMC after the treatment and the ASE after each wet-dry cycle. The EMC was 4.5%–6.3% in the chemically treated groups, which was slightly lower than the heat-treated group H. The ASE after the first cycle was high among all groups and related to the WPG. Alongside the volatile extractives, nonvolatile degradation products were formed during thermal degradation, which remain in the wood and bulk the cell wall [7]. However, after three cycles, the ASE decreased significantly, indicating that the chemicals and or water-soluble degradation products were leached out. This effect was strongest in the succinimide groups and lowest in the CIOL® groups. The low ASE for succinimide supports the theory that no modification of the wood took place. After three cycles, the ASE of the I and Su groups were not related to the initial WPG, whereas all groups containing citric acid were related to WPG. The ASE of I10 was 33.4%, and the addition of citric acid (ICA10) resulted in an ASE of 34.7%, indicating no synergistic effect of imidazole and citric acid. However, in the case of succinimide, the ASE increased from 21.0% to 37.9% via the addition of citric acid. For CIOL®, the high ASE is attributed to the formation of the insoluble polymers formed during the esterification process between citric acid and wood and citric and sorbitol [30]. Furthermore, the results are in line with work from Beck [35].

Figure 4 presents the mass loss caused by wet-dry cycling, i.e., the leaching of chemicals, degraded wood components, and water-soluble extractives, and the mass loss over the whole treatment and leaching procedure. $\Delta m_{leaching}$ correlated with the WPG in each group except for CIOL®. Combinations, including citric acid, exhibited less mass loss than the respective single compound treatments. As shown in Figure 2, different degrees of mass loss occurred during the heat treatment, which influenced $\Delta m_{leaching}$. In order to understand the mode of action of the treatments, it was, thus, more conclusive to study the total mass loss $\Delta m_{total}$. It was noted that $\Delta m_{leaching}$ in the imidazole and succinimide groups was rather similar, but the values for $\Delta m_{total}$ were significantly different. Treatments involving imidazole groups exhibited a mass loss of 7%–8%, indicating that the wood components were degraded to an extended degree during the heat treatment and were washed out during the wet-dry cycling; this was supported by the results for heat treatment only, where the results showed a mass loss of 3%. The yellow/brown color of the leachate supported this.

**Table 2.** The mean equilibrium moisture content (EMC) and anti-swelling efficiency (ASE) of specimens treated with different concentrations (wt%) of the chemical solution followed by heat treatment. Number of specimens per group = 6.

| | I | | | ICA | | | Su | | | SuCA | | | CIOL® | | | H | C |
|---|---|---|---|---|---|---|---|---|---|---|---|---|---|---|---|---|---|
| **(wt%)** | **5** | **10** | **15** | **5** | **10** | **15** | **5** | **10** | **15** | **5** | **10** | **15** | **5** | **10** | **15** | **-** | **-** |
| EMC | 4.5 | 5.2 | 4.6 | 5.2 | 5.3 | 5.0 | 6.3 | 6.1 | 5.6 | 5.3 | 6.1 | 4.8 | 5.5 | 6.3 | 5.0 | 7.0 | 11.0 |
| ASE 1st cycle | 43.7 | 53.8 | 59.8 | 38.0 | 44.1 | 54.4 | 36.3 | 47.2 | 57.4 | 45.6 | 50.6 | 59.2 | 40.8 | 47.9 | 52.7 | 22.4 | 0 |
| ASE 2nd cycle | 34.5 | 35.2 | 37.1 | 32.7 | 37.0 | 43.4 | 22.7 | 24.2 | 27.2 | 34.2 | 40.5 | 46.5 | 36.4 | 43.4 | 49.9 | 14.7 | −8.1 |
| ASE 3rd cycle | 29.6 | 33.4 | 31.3 | 29.2 | 34.7 | 38.2 | 19.0 | 21.0 | 17.4 | 31.6 | 37.9 | 41.2 | 35.0 | 41.5 | 47.5 | 11.1 | −7.0 |

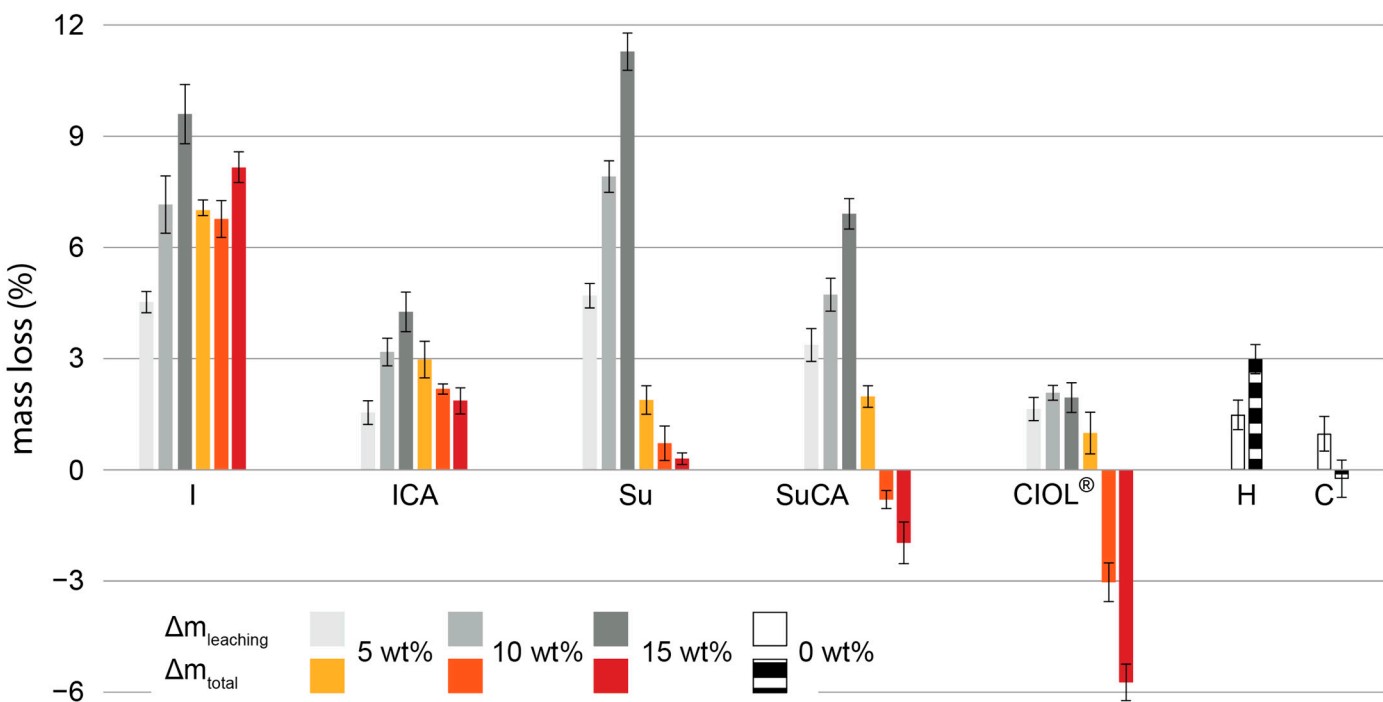

**Figure 4.** Mass loss after three wet-dry cycles in reference to the dry weight after the treatment ($\Delta m_{leaching}$) and to the initial dry weight ($\Delta m_{total}$). wt% corresponds to the solid concentration of the chemical solution used for pressure impregnation. The number of specimens per group = 6.

The WPG did not have a strong influence on the $\Delta m_{total}$, which was in line with the ASE values. The reductions in ASE from 50–60% in the first cycle to around 30% in the third cycle, coupled with a $\Delta m_{total}$ of 7%–8% were similar to the results of thermal modification by Wentzel et al. [51]. However, if imidazole would solely promote thermal degradation, the mechanical properties would be expected to decrease, which was not the case. The addition of citric acid presented in the ICA groups drastically lowered $\Delta m_{total}$ due to a lower concentration of imidazole and the presence of citric acid. It is also possible that citric acid reacted with the degradation products of the imidazole-promoted reactions, resulting in insoluble polymers and a low $\Delta m_{leaching}$ and $\Delta m_{total}$. Succinimide showed the highest $\Delta m_{leaching}$ but a $\Delta m_{total}$ close to zero. When taken together with the low and WPG-independent $\Delta m_{HT}$ and ASE, this indicated that succinimide either did not react with the wood components or only a few reaction sides were available. The addition of citric acid (SuCA) improved leaching resistance as well as $\Delta m_{total}$. The negative mass loss indicated that insoluble polymers were formed. It is also possible that a reaction of succinimide with

the wood or the citric acid took place due to the more acidic conditions. CIOL® treatment led to little leaching and higher chemical retention due to the formation of polymers in the esterification process, and this has been shown by previous research by Larnøy et al. [30].

Figure 5 shows the volumetric change over the wet-dry cycles of the specimens treated with 15 wt% solutions, with D0 being equivalent to the BC. The chemical treatments led to a relatively low BC, with the exception of the succinimide treatments. However, except for CIOL®, the cell wall bulking reduced during the cycles as the chemicals and degradation products were leached out. The reduction in volume was smaller in the groups containing citric acid. Treatment with imidazole alone led to the smallest dimensions after three cycles, assumed to be due to the degradation of lignin. Groups containing citric acid exhibited higher dimensional stability, indicating some degree of crosslinking in ICA and SuCA. Whether these crosslinks involve imidazole or succinimide is currently uncertain. Succinimide led to a reduction in shrinking but not swelling in comparison to the unmodified wood, indicating that although cell wall bulking occurred, there was no crosslinking.

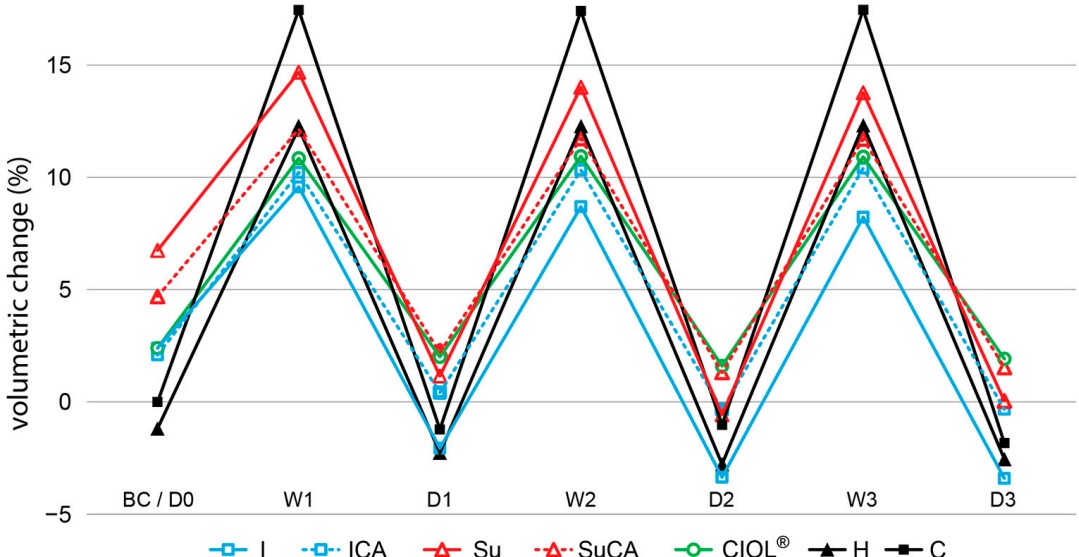

**Figure 5.** Mean volumetric changes in specimens treated with solutions with 15 wt% solid concentration during wet-dry cycling. Dn: oven-dried in the nth cycle; Wn: wet in the nth cycle.

### 3.3. Bending Performance

MOR and MOE are presented in Figure 6. The statistical analysis suggested that little significant differences existed between the groups. However, the large span of initial density in each group as a result of the even group distribution and natural variability in wood might be a reason for this, as the mean values exhibited clear differences. The mean MOR of some groups was slightly higher than the untreated control, which might be caused by a reduction in moisture content due to thermal treatment. The I and ICA groups showed high MOR at 5 and 10 wt%, with a reduction in MOR at 15 wt%. The increased degradation of cell wall polymers was likely concentrated in the lignin, as holocellulose degradation is expected to result in greater reductions in mechanical properties due to increased shear slipping between cellulose microfibrils [52]. The SuCA and CIOL® groups showed a reduction in MOR. For CIOL®, this is attributed to the acidic modification conditions that led to thermally induced mechanical degradation [34]. A similar drop in the MOR was found in SuCA but not in ICA. Interestingly, Su5 showed the highest MOR, and Su15 had one of the lowest MORs. It is possible that the combined succinimide and citric acid treatment resulted in a similar formation of crosslinks as in the CIOL® process. The MOE was less affected by the treatments. Except for the succinimide groups, the MOE seemed to be unaffected by WPG, and the slight increase in MOE is likely due to the heat treatment.

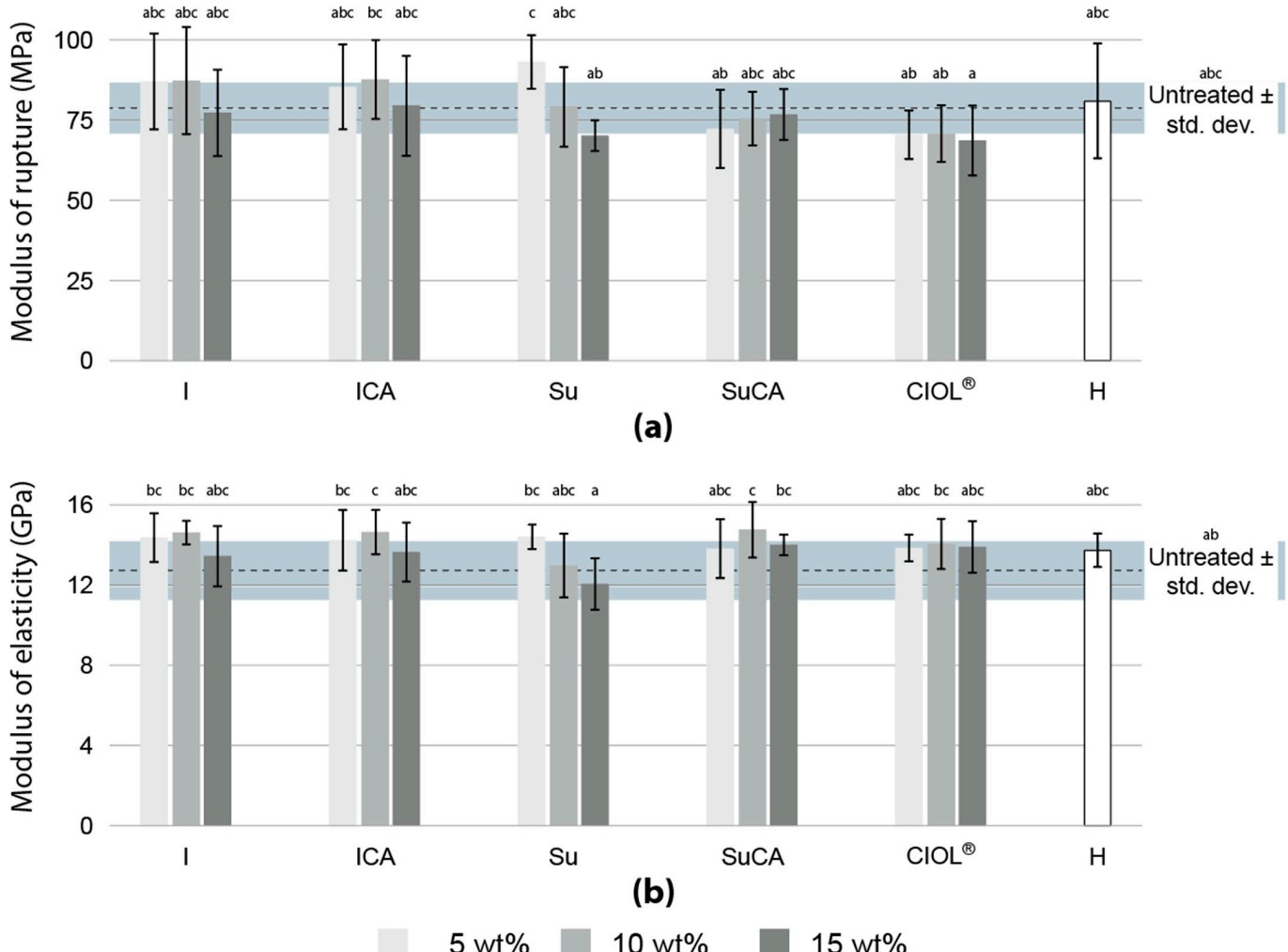

**Figure 6.** The mean ± standard deviation of the modulus of rupture (**a**) and modulus of elasticity (**b**) were measured for the untreated and treated specimens. wt% corresponds to the solid concentration of the chemical solution used for pressure impregnation. The result of the statistical post-hoc test is presented by the compact letter display above each bar. The number of specimens per group = 12.

The results show that the wood modification with imidazole does not negatively affect bending performance but does provide increased dimensional stability. Both the positive effect on ASE and the negative effect on MOR, are stronger in the succinimide and CIOL® treatments. In-house experience that was not included in this study indicated that some degree of cross-polymerization occurs in I, ICA, and SuCA, with ongoing work to provide definitive answers.

## 4. Conclusions

In order to facilitate the use of fossil-free modification reagents, this study aimed to explore wood modification systems involving imidazole and succinimide in combination with citric acid. The treatments were based on pressure impregnation and subsequent heat treatment to improve the hygroscopic properties without aggravating the mechanical performance.

The treatments with imidazole exhibited increased mass loss during heat treatment, which led to the formation of water-soluble degradation products that were leached out over the wet-dry cycles. The mass loss during the heat treatment of succinimide-containing treatments seemed to be unaffected by the chemical, and a large amount of succinimide was leachable. The total mass loss from the untreated state to the leached state of the

imidazole-treated specimens was 7%–8%, exceeding the sole heat treatment mass loss of 3%. The succinimide treatment exhibited values of 0%–2%, indicating a degree of chemical retention. Substituting a part of the chemicals with citric acid improved the leaching resistance and decreased the total mass loss. It is not clear if this was solely due to the reaction of citric acid with the cell wall polymers, or if crosslinks were formed between citric acid and imidazole or succinimide. The reactions between citric acid and the imidazole-promoted lignin-derived degradation products are another possibility. Leaching resistance was highest for a combination of citric acid and sorbitol (CIOL®-process). However, in this treatment, the share of citric acid was almost twice as high as in the other treatments containing citric acid. After three cycles, the anti-swelling efficiency (ASE) reached 31% for the imidazole-treated specimens and improved to 38% with the addition of citric acid. For succinimide, the ASE increased from 17% to 41%. CIOL® exhibited an ASE of 48%.

Previous research showed that wood modification with citric acid led to a high ASE and high leaching resistance at the cost of strongly reduced mechanical properties. In the current study, the bending properties generally showed an improvement, except for succinimide at a high concentration, succinimide + citric acid, and CIOL®, which displayed a reduced modulus of rupture. This indicates that wood modification with imidazole and succinimide are based on different reaction mechanisms, where the improved hygroscopic properties of imidazole treatment does not solely originate from an enhanced thermal degradation, as the modulus of rupture should be strongly reduced. In contrast, succinimide and citric acid might polymerize with each other and any residual wood components, which is akin to the CIOL® process, as the treatments exhibited a similar reduction in modulus of rupture.

The results of this study show that wood modification with imidazole and succinimide can be utilized. However, further research should delve into analyses of the involved reaction mechanisms to optimize the chemical ratios and leaching resistance. Studies on the impact of imidazole and succinimide on biological durability are currently ongoing.

**Author Contributions:** Conceptualization, D.J., D.S. and H.D.; methodology, D.J.; validation, D.S., H.D. and J.O.; formal analysis, A.S.; investigation, A.S. and D.J.; data curation, A.S.; writing—original draft preparation, A.S.; writing—review and editing, A.S., D.J. and D.S.; visualization, A.S.; supervision, D.J.; project administration, D.J. and D.S.; funding acquisition, D.J. and D.S. All authors have read and agreed to the published version of the manuscript.

**Funding:** Support through the project CT WOOD, a center of excellence at Luleå University of Technology, and the VINNOVA project "Multifunktionella byggskivor av sågspån" (Grant no. 2022-00998) is gratefully acknowledged.

**Data Availability Statement:** The data presented in this study are available on request from the corresponding author. All data is supported in the paper (with statistical variance).

**Conflicts of Interest:** The authors declare no conflict of interest. The funders had no role in the design of the study; in the collection, analyses, or interpretation of data; in the writing of the manuscript; or in the decision to publish the results.

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
