# Peer review of "Wood Modification Using Imidazole and Succinimide: Effects on Dimensional Stability and Bending Properties"

_forests, doi:10.3390/f14101976_

Round 1

Reviewer 1 Report

The study focused on modifying pine wood using imidazole and succinimide compounds. Results indicated that concentrations between 5% and 10% were most effective. Imidazole resulted in a higher mass loss due to heat treatment, while succinimide showed stability. The addition of citric acid improved the Anti-Swelling Efficiency (ASE). Bending properties generally improved, except for specific combinations. Further chemical analyses are needed for a complete understanding of the processes. Please see below for my comments on your manuscript:

Materials Description:

Specify the exact source of the Scots pine (Pinus sylvestris L.) sawn timber in Sweden (e.g., coordinates).

Include more details about the industrial drying process, such as the duration and temperature used to reach 18% moisture content (MC). This information helps in replicating the experiment.

Chemicals Used:

Clearly state the purity levels of the chemicals (e.g., imidazole, succinimide, citric acid, and sorbitol).

Mention the specific grades or specifications of analytical-grade citric acid and technical-grade D(-)-sorbitol.

Provide references or certificates for the purity of the chemicals to ensure reproducibility.

Specimen Preparation:

Clarify the method used for selecting defect-free, straight-grained sapwood specimens from the timber. Was it done visually, using any specific criteria?

Specify the equipment and techniques used for conditioning the specimens to EMC. Include the duration and methods used to achieve this equilibrium.

Pressure Impregnation and Heat Treatment:

Provide a more detailed description of the pressure impregnation process, including the type of vessel, pressure control method, and any safety precautions taken.

Clarify the significance of the vacuum stage in the impregnation process.

Elaborate on the assumption that little to no chemical reaction occurred during drying and provide any evidence supporting this claim.

Specimen Characterization:

Offer more information on the wet-dry cycle process, including the reason for three cycles and any specific testing standards followed.

Explain the relevance of evaluating mass loss due to leaching and the implications of this measurement.

 The "Results and Discussion" section is crucial in presenting the findings of the study and discussing their implications:

Impregnation and Heat Treatment:

The description of solution uptake and WPG is clear. However, consider providing the numerical values for these percentages, which would enhance the clarity of the results.

The description of WPG behavior is concise, but you could provide more insights into the reasons behind the observed trends. For instance, explain why WPG doubled with a 5% to 10% solid concentration but did not similarly increase from 10% to 15%.

Consider discussing any potential practical implications of these findings. For example, how might these WPG variations affect the wood's properties or applications?

When discussing the chemical treatment groups, emphasize the practical implications of these findings. For instance, how might the presence of imidazole affect wood degradation during heat treatment, and how could this impact practical applications?

Bending Performance:

The presentation of MOR and MOE is informative, but it could benefit from a more detailed discussion of the results.

Clarify the significance of the MOR (modulus of rupture) and MOE (modulus of elasticity) values in the context of the study. What do these mechanical properties indicate about the wood's performance?

While it's mentioned that there were no significant differences between groups, it's essential to interpret these results. Why were there no significant differences, and what can be inferred from this lack of statistical significance?

Author Response

Many thanks for taking the time to review this article and your thoughtful comments. Please see the attached document for the direct response to these.

Kind regards

Dennis Jones

Reviewer 2 Report

Comments and Suggestions for Authors

Dear Authors and Editors,

The results of this research are significant, and all conclusions are justified and supported by the results.

The paper presents original research work and provides an advance in current knowledge.

Extensive and future promising research was conducted with important insight into the wood modification with imidazole and succinimide. This is the first test of wood modification with the chemicals described at the conclusion of the introduction and it yielded several novel discoveries.

The authors have conducted a good study and explained the importance of their research. The methodology is correct, and the results are well explained.

English language and style are acceptable.

The paper is acceptable for publishing in the journal Forests after minor corrections.

General questions:

Would additional characterizations using FTIR, TG, and SEM analysis be reasonable to gain more information?

Why was scot pine the first sort of wood to undergo the aforementioned modification? Given that it is such a detailed job I believe it would be beneficial to write a few sentences about it.

Line

Line 127: Would using natural drying be preferable?

Line 144: A brief explanation of silicone's purpose is provided in parentheses.

Author Response

(The authors gave the same response as above.)

Reviewer 3 Report

In this work, imidazole and succinimide was selected for wood modification. WPG, ASE, colorimetry, bending properties were investigated. All results and conclusions give some practices advice for modify wood research. However, there also have many questions as follows should be addressed.

1) What is the aim for thermal treatment combined with chemical modification? The treatment with chemistry (imidazole, succinimide or critic acid) lonely is required as control.

2) After impregnation, wood tissue was more easily degradation during thermal treatment and result in increased mass loss with increase of chemistry concentration. However, why there isn’t decreasing of mass loss when treated with succinimide at 10%?

3) When treated with imidazole added critic acid at 10%, there is not showing increased WPG and decreased mass loss. However, it was happened in treatment of succinimide with critic acid. Whether it is accurately caused by the esterification in critic acid and chemistry or wood cell walls?

4) Does the dehydration in wood cell walls will be increasing under 220 oC thermal treatment combined with critic acid impregnation?

5) With more recycles, heavily WPG modified wood showed obviously decrease of ASE. However, there is not present in treatment by critic acid with sorbitol. As a result, modified wood by higher concentration of imidazole or succinimide will be deposited into wood lumen or intercellular space. And when in certain condition this deposition will leaching which might be aggravated due to degradation of wood tissue or physical absorption.

6) For Scots pine, the MOR generally is 80-90MPa and MOE is 10-12GPa. But in this work, all modified wood processed in thermal condition and after impregnation with chemistry reagents result in 5%-15% mass loss. Based on the experiment modified wood showed questionable MOR about 85 MPa and MOE nearly 14GPa.

Author Response

Many thanks for taking the time to review this paper and for your thoughtful comments. Please see the attached document for our response to these.

Kind regards

Dennis Jones

Round 2

Reviewer 3 Report

No any questions and adivsed been accepted for this editor.

No any questions and adivsed been accepted for this editor.